# Ground-state correlation energy
# of beryllium dimer by the Bethe-Salpeter equation

Jing Li[1,2,3], Ivan Duchemin[3] Xavier Blase[1,2] and Valerio Olevano[1,2,4]

**1** Université Grenoble Alpes, 38000 Grenoble, France
**2** CNRS, Institut Néel, 38042 Grenoble, France
**3** Univ. Grenoble-Alpes, CEA, L_Sim, 38000 Grenoble, France
**4** European Theoretical Spectroscopy Facility (ETSF)

## Abstract

Since the '30s the interatomic potential of the beryllium dimer $Be_2$ has been both an experimental and a theoretical challenge. Calculating the ground-state correlation energy of $Be_2$ along its dissociation path is a difficult problem for theory. We present *ab initio* many-body perturbation theory calculations of the $Be_2$ interatomic potential using the *GW* approximation and the Bethe-Salpeter equation (BSE). The ground-state correlation energy is calculated by the trace formula with checks against the adiabatic-connection fluctuation-dissipation theorem formula. We show that inclusion of *GW* corrections already improves the energy even at the level of the random-phase approximation. At the level of the BSE on top of the *GW* approximation, our calculation is in surprising agreement with the most accurate theories and with experiment. It even reproduces an experimentally observed flattening of the interatomic potential due to a delicate correlations balance from a competition between covalent and van der Waals bonding.

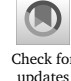
# 1 Introduction

The beryllium dimer $Be_2$ has a long scientific history, with hundreds of experimental and theoretical investigations [1, 2]. The first synthesis of $Be_2$ was attempted in the '30s with no success [3, 4], while Hartree-Fock (HF) or other theoretical modelling [5, 6] found a repulsive ground-state, leading to the conclusion that $Be_2$ does not exist. Later studies [2, 7] pointed to a possible van der Waals binding with a shallow energy minimum at large ($\sim 5$ Å) distance, while other studies yielded a double minimum, at short and long distance separation [2, 8]. $Be_2$ remained elusive till the '70s [9], and only in the '80s first rotovibrational spectra were measured [10] and reliable calculations [11] were made, both pointing to a single short-bond minimum at $\sim 2.5$ Å. Today we have very accurate experiments [1] and calculations [12] for the $Be_2$ interatomic potential. Nevertheless, $Be_2$ remains a severe workbench to check many-body theories, and this is the purpose of this work.

The description of the correlation energy of molecular dimers along their dissociation path is a theoretical challenge [13, 14]. Full configuration interaction (CI) [14] is the only accurate method [2, 12] providing results in very good agreement with experiment, but it is limited to very small molecules. Quantum Monte Carlo (QMC) [13, 15–19], both variational (VMC) and diffusion (DMC) are also valid alternatives, and DMC is even exact in systems where the ground-state wavefunction does not present nodes, but they can be also very cumbersome. Density-functional theory (DFT) [13] is in principle exact for calculating ground-state energies, but standard approximations, like the local-density (LDA) and generalized-gradient (GGA) approximations, have shown their limits on dimers binding energies and lengths [20], in particular in the dissociation limit. In recent years time-dependent density-functional theory (TDDFT) [21, 22] in the adiabatic-connection fluctuation-dissipation theorem (ACFDT) formalism [23, 24] has been considered as one promising approach to improve over approximated DFT [20, 25–27]. In any case it relies on TDDFT approximations for the polarizability $\chi$, such as the random-phase approximation (RPA), the adiabatic LDA (ALDA or TDLDA), or beyond. Nevertheless, the interatomic potential of $Be_2$ continues to be a problem also for TDDFT ACFDT, both in the RPA [28] and also more advanced approximations [27, 29].

In this work, we calculate the $Be_2$ interatomic potential in the framework of *ab initio* many-body perturbation theory using the *GW* approximation [30] and the Bethe-Salpeter equation (BSE) [13, 31, 32]. The ground-state correlation energy is calculated by the trace formula (TF) [33, 34]

$$E_0^c = \frac{1}{2}\left[\sum_{i>0}\Omega_i - \text{Tr}(A)\right],\tag{1}$$

where $\Omega_i$ are the positive eigenvalues of the full, i.e. beyond the Tamm-Dancoff approximation, BSE equation, and $A$ is the only-resonant part of the BSE excitonic Hamiltonian. This formula was introduced by Sawada to calculate the correlation energy of the electron gas by summing over the contributions from zero-point plasma oscillations (plasmons) [35] as an alternative to the Gell-Mann & Brueckner formula which integrates along the adiabatic connection path over the interaction switch-on $\lambda$ parameter [36]. Later these two formulae were shown to be equivalent within RPA, both on the electron gas [37, 38] and for inhomogeneous systems in using the DFT Kohn-Sham (KS) as reference [39]. This allows us to validate our results from Eq. (1), at least at the RPA level, by also evaluating the correlation energy via the RPA ACFDT formula starting from KS eigenstates [20, 23, 24]

$$E_0^c = \frac{1}{2\pi}\int_0^{i\infty} d\omega \, \text{Tr}\left[\ln\left(1 - w\chi^0(\omega)\right) + w\chi^0(\omega)\right].\tag{2}$$

Here $w(r, r') = 1/|r - r'|$ is the Coulomb interaction, $\chi^0(r, r', \omega)$ is the Kohn-Sham polarizability, and the integral over $\omega$ is along the positive imaginary axis. Our results show that

*GW* corrections introduce large improvements already at the level of the RPA. The interatomic potential we obtain at the level of the BSE on top of the *GW* approximation is in surprising agreement with the most accurate calculations and experiment. *GW*+BSE correlations even seem able to describe the unusual shape of the experimental [1] interatomic potential, namely a flattening of the Morse potential in the range between 6 and 9 bohr (3 to 4.5 Å) towards an expanded Morse potential which better fits the experimental vibrational spectrum [1].

The solution of the historical problem represented by the paradigmatic $Be_2$ interatomic potential shows that the use of Eq. (1) in the BSE framework can reveal an accurate methodology for calculating the ground-state energy and stability of atoms and molecules, solids, and even nuclei, with an important advance in all these fields.

## 2   Theory

Eq. (1) was derived previously in different ways [35, 38, 39]. Here we present a modification of the Thouless derivation [33, 40] which extends its validity to starting points different from HF towards DFT or *GW*, and to kernels beyond RPA towards BSE or TDDFT kernels. We start from the Bethe-Salpeter equation,

$$L = L^0 + L^0 \Xi L, \tag{3}$$

where $L$ is the two-particle correlation function, $L^0 = GG$ with $G$ the one-particle Green function, and $\Xi$ the two-particle interaction. This is an exact equation for calculating the excitation spectrum. By knowing the exact $\Xi$ and $L^0$ (via $G$), the BSE can be solved for $L$ whose poles $\Omega_\lambda = E_\lambda - E_0$ and their associated residuals provide the neutral (optical) excitation energies and oscillator strengths. In practice, approximations are unavoidable. We consider a quite general case of an approximated electronic structure, e.g. HF, *GW* or DFT, with real energies $\epsilon_i$ and orthonormal wavefunctions $\phi_i(r)$ used to build $G$ and $L^0$. And we consider an approximated static kernel, e.g. the kernel TDH $i\Xi_{ijkl} = w_{iljk}$ (with $w$ the bare Coulomb interaction), or the TDHF $i\Xi_{ijkl} = w_{iljk} - w_{ilkj}$, or the *GW*+BSE $i\Xi_{ijkl} = w_{iljk} - W_{ilkj}$ (with $W$ the screened Coulomb interaction), or even the TDLDA. Under these conditions the Bethe-Salpeter equation can be reduced to the well known [13, 33, 41–43] RPA equation

$$\begin{pmatrix} A & B \\ B^* & A^* \end{pmatrix} \begin{pmatrix} X^\lambda \\ Y^\lambda \end{pmatrix} = \Omega_\lambda \begin{pmatrix} 1 & 0 \\ 0 & -1 \end{pmatrix} \begin{pmatrix} X^\lambda \\ Y^\lambda \end{pmatrix}, \tag{4}$$

with the Hermitian matrix $A_{tt'} = A_{php'h'} = (\epsilon_p - \epsilon_h)\delta_{pp'}\delta_{hh'} + i\Xi_{php'h'}$ and the symmetric matrix $B_{tt'} = B_{php'h'} = -i\Xi_{pp'hh'}$ indexed by particle-hole transition indices $t = \{ph\}$ from an occupied state $h$ to an empty state $p$, $\hat{c}_p^\dagger \hat{c}_h$, on the orthonormal basis set $\phi_i(r)$. Eq. (4) provides a full spectrum of excitations $|\Psi_\lambda\rangle$, both the excitation energies $\Omega_\lambda = E_\lambda - E_0$ (with respect to the ground state energy $E_0$) and the eigenvectors $(X^\lambda Y^\lambda)$. This spectrum constitutes an approximation to the exact spectrum. We now introduce *boson* [41] transition operators $\hat{C}_t$ replacing the *ph* operator bilinears, $\hat{c}_p^\dagger \hat{c}_h \to \hat{C}_t^\dagger$, i.e. fulfilling exact boson canonical commutation relations, $[\hat{C}_t, \hat{C}_{t'}] = 0, [\hat{C}_t^\dagger, \hat{C}_{t'}^\dagger] = 0, [\hat{C}_t, \hat{C}_{t'}^\dagger] = \delta_{tt'}$. This is not the case for the fermion bilinears, $[\hat{c}_p \hat{c}_h^\dagger, \hat{c}_p^\dagger \hat{c}_h] \neq \delta_{pp'}\delta_{hh'}$. So the boson operators can be seen as an approximation to the fermion bilinears. We now express the full many-body Hamiltonian $\hat{H}$ in terms of the boson operators imposing the condition that we get, by construction, the same excitation spectrum of Eq. (4). This is achieved if [41]

$$\begin{aligned} \langle\Psi_0|[\hat{C}_t, [\hat{H}, \hat{C}_{t'}^\dagger]]|\Psi_0\rangle &= A_{tt'}, \\ \langle\Psi_0|[\hat{C}_t, [\hat{H}, \hat{C}_{t'}]]|\Psi_0\rangle &= -B_{tt'}. \end{aligned}$$

Up to quadratic terms only [44] the Hamiltonian is written

$$\hat{H} = E_0^0 + \sum_{tt'} A_{tt'} \hat{C}_t^\dagger \hat{C}_{t'} + \frac{1}{2} \sum_{tt'} [B_{tt'} \hat{C}_t^\dagger \hat{C}_{t'}^\dagger + B_{tt'}^* \hat{C}_t \hat{C}_{t'}],$$

where the constant $E_0^0$ is approximated by the expectation value of the Hamiltonian, $E_0^0 = \langle \Phi_0 | \hat{H} | \Phi_0 \rangle$, over the zero-order ground-state Slater determinant $\Phi_0$ constructed with the occupied wavefunctions $\phi_h(r)$ [1]. For HF orbitals $\phi^{\mathrm{HF}}(r)$, the constant is the HF ground-state energy $E_0^0 = E_0^{\mathrm{HF}}$, whereas in the case of DFT the constant is the sum of the kinetic, external, Hartree and exchange operators evaluated on the Kohn-Sham wavefunctions, $E_0^0 = E^{\mathrm{DFT}} - E_{\mathrm{xc}} + E_{\mathrm{EXX}}$. The Hamiltonian can be recast into a matrix form,

$$\hat{H} = E_0^0 - \frac{1}{2}\mathrm{Tr}(A) + \frac{1}{2} \begin{pmatrix} \hat{C}^\dagger & \hat{C} \end{pmatrix} \begin{pmatrix} A & B \\ B^* & A^* \end{pmatrix} \begin{pmatrix} \hat{C} \\ \hat{C}^\dagger \end{pmatrix},$$

and the solutions $(X^\lambda \, Y^\lambda)$ to Eq. (4), subject to the orthonormality condition $\sum_t (X_t^{\lambda*} X_t^{\lambda'} - Y_t^{\lambda*} Y_t^{\lambda'}) = \delta_{\lambda\lambda'}$, allows us to define new boson operators $\hat{Q}_\lambda$ by a Bogoliubov transformation of the $\hat{C}_t$,

$$\hat{Q}_\lambda^\dagger = \sum_t (X_t^\lambda \hat{C}_t^\dagger - Y_t^\lambda \hat{C}_t),$$

which diagonalizes the Hamiltonian

$$\hat{H} = E_0^0 - \frac{1}{2}\mathrm{Tr}(A) + \frac{1}{2} \sum_\lambda \Omega_\lambda + \sum_\lambda \Omega_\lambda \hat{Q}_\lambda^\dagger \hat{Q}_\lambda. \tag{5}$$

The operators $\hat{Q}_\lambda$ act on the (approximated) ground $|\Psi_0\rangle$ and excited $|\Psi_\lambda\rangle$ states, $\hat{Q}_\lambda^\dagger |\Psi_0\rangle = |\Psi_\lambda\rangle$, $\hat{Q}_\lambda |\Psi_0\rangle = 0$. And the expectation value of the Hamiltonian Eq. (5) over the excited states $|\Psi_\lambda\rangle$ provides, by construction, the excitation energies $\Omega_\lambda$ with respect to the ground state. Finally, the expectation value of the Hamiltonian Eq. (5) over $|\Psi_0\rangle$ provides the total energy of the ground state *including correlation* (within the stated approximations),

$$E_0 = E_0^0 + \frac{1}{2} \sum_\lambda \Omega_\lambda - \frac{1}{2}\mathrm{Tr}(A). \tag{6}$$

Thus, for a starting HF electronic structure, the two terms after the constant $E_0^0 = E_0^{\mathrm{HF}}$ in Eq. (6) provide the correlation energy Eq. (1), alternatively recast in the form

$$E_0^{\mathrm{corr}} = \frac{1}{2} \sum_\lambda (\Omega_\lambda - \Omega_\lambda^{\mathrm{TDA}}), \tag{7}$$

or in the form

$$E_0^{\mathrm{corr}} = -\sum_\lambda \Omega_\lambda \sum_t |Y_t^\lambda|^2, \tag{8}$$

where $\Omega_\lambda^{\mathrm{TDA}}$ are the eigenvalues of Eq. (4) in the Tamm-Dancoff approximation (TDA), i.e. taking the matrix $B = 0$. Eqs. (7) and (8) show more clearly that the ground state correlation energy arises only beyond the TDA approximation and from terms in $B$ and $Y$. The TDA is only able to introduce correlations in the excited states.

To summarize, Eq. (6) is an expression for the ground-state total energy $E_0$ *including correlation at the same level of approximation* (e.g. RPA or TDHF, $GW$+BSE, etc.) as taken for the

---

[1]More exactly $E_0^0 = \langle \Psi_0^0 | \hat{H} | \Phi_0^0 \rangle$ where $\Psi_0^0$ is the ground-state killed by the boson operators, $\hat{C}_t |\Psi_0^0\rangle = 0$. Since $\hat{C}_t \simeq \hat{c}_p^\dagger \hat{c}_h$, the 0-order ground state $\Phi_0$, killed by the $\hat{c}_p^\dagger \hat{c}_h |\Phi_0\rangle = 0$, is an approximation for $\Psi_0^0$, and $\hat{C}_t |\Phi_0\rangle \simeq 0$ is approximated to zero.

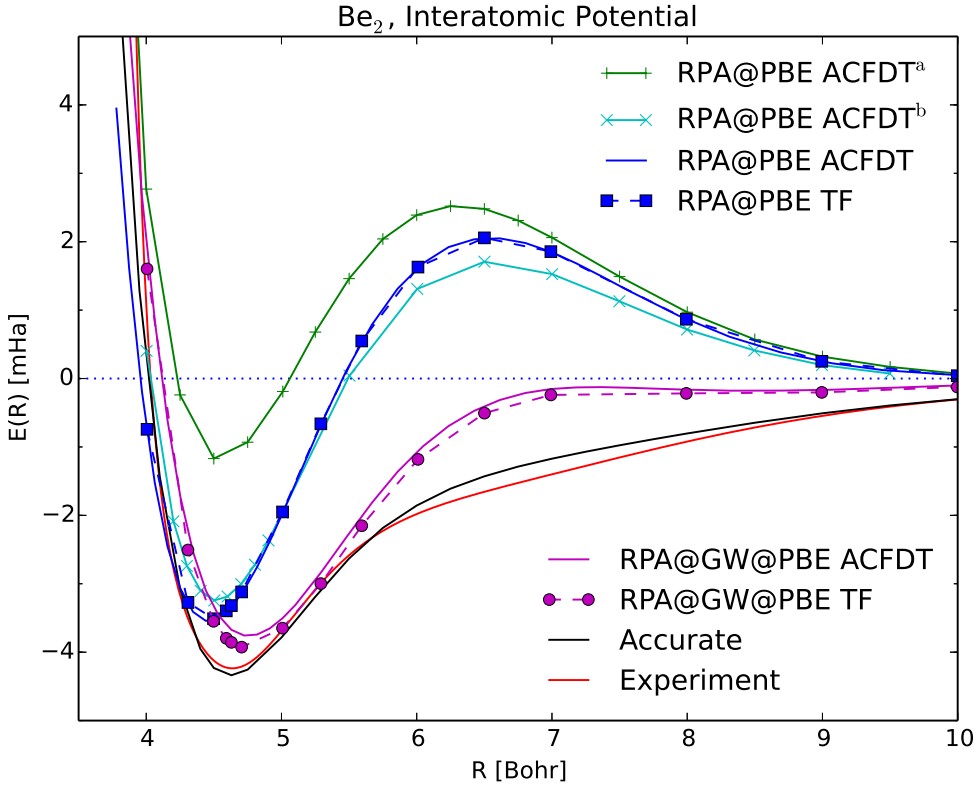

Figure 1: The interatomic potential of $Be_2$. Red line: experiment [1]; black line: accurate CI [12]; green line with plus symbols (a): RPA on top of PBE by ACFDT from Ref. [29]; cyan line with cross symbols (b): RPA on top of PBE by ACFDT from Ref. [28]; blue continuous line: RPA on top of PBE by ACFDT, our calculation; blue dashed line with squares: RPA on top of PBE by the trace formula, Eq. (6); magenta continuous line: RPA on top of $GW$ and PBE by ACFDT; magenta dashed line with circles: RPA on top of $GW$ and PBE by TF.

$A$ and $B$ matrices, and so at the same level of approximation of the associated excitation spectrum $\Omega_\lambda$. In contrast to the ACFDT formalism, which provides an in principle exact formula for correlations within TDDFT, Eq. (6) is only an approximate expression since the beginning that relies on the validity of the boson approximation between operators, $\hat{C}_t \simeq \hat{c}_p^\dagger \hat{c}_h$, and the validity of the killing condition on the zero-order Slater determinant ground-state, $\hat{C}_t|\Phi_0\rangle \simeq 0$. However for both formulae most critical are the approximations on the kernel (RPA and beyond) and on the starting electronic structure (LDA or else). Comparing the merits of these various approximations is still in its infancy for atoms and molecules [45, 46].

In order to provide the best comparison with previous literature and with ACFDT results, we use the same large cc-pV5Z Gaussian basis set [47] adopted in Ref. [29], together with the auxiliary cc-pV5Z-RI basis set [48] in a Coulomb-fitting, resolution-of-identity approach. Input Kohn-Sham or Hartree-Fock eigenstates were calculated by the NWCHEM [49] package, whereas many-body calculations were performed with the FIESTA [50, 51] code. For improved accuracy [50, 52] we performed ev$GW$ calculations, namely partially self-consistent $GW$ on the eigenvalues only. $GW$ corrections were calculated explicitly for all 4 occupied and 14 empty states, while corrections to higher states were extrapolated by a rigid scissor-operator shift from the last calculated level. This provided a result converged up to 0.2 mHa.

# 3 Results

In Fig. 1 we present the interatomic potential of $Be_2$ as derived from recent accurate experimental rotovibrational spectra [1], and from the most accurate configuration interaction (CI) calculation [12] which is in good agreement with the experiments. On the same figure, we present the theoretical curve that we calculated at the level of the direct RPA approximation on top of DFT in the PBE [53] approximation (RPA@PBE) by the TF formula Eq. (6) or equivalently Eq. (8) which, we checked, provide the same result within numerical precision. We show also the same RPA@PBE curve calculated by the ACFDT formula. They are in perfect agreement. We can compare our RPA@PBE curves with those calculated by other authors [28, 29] with the ACFDT formula. Although we used exactly the same calculation parameters reported in Ref. [29], our result differ from theirs, in particular for the binding energy, due to the basis set superposition error that they mentioned and which they removed by counterpoise. We prefer not to use the counterpoise method because some works [54, 55] showed it is not justified in the case of systems where van der Waals interaction can be important. For this reason our result is closer, surprisingly, to the one of Ref. [28] obtained by a much different implementation, namely plane-waves and norm-conserving pseudopotentials. Nevertheless, all RPA@PBE calculations present qualitatively the same fake feature, a "bump" (a maximum) at 6–7 Bohr, in contrast to experiment and to the accurate CI calculation. Note that the bump is completely absent in the original DFT PBE interatomic potential which is strongly attractive everywhere with a deep minimum and large binding energy (see Ref. [27]). Finally, we present curves calculated at the same level of direct RPA but using a *GW* corrections on top of PBE. The most important point is that there is a large improvement in RPA@*GW*@PBE with respect to RPA@PBE. The "bump" is wiped out. The binding energy is also improved, though still underestimated. The bonding length is now overestimated.

The bump at 6-7 Bohr appears in TF or ACFDT RPA calculations on top of DFT PBE (as well as LDA, see Ref. [28]) but it is conjured by the exact exchange (EXX) term evaluated on DFT PBE eigenstates (but not HF) and wiped out by a *GW* calculation with self-consistency on the eigenvalues only. This can be appreciated also in Fig. 2 where we present an RPA calculation on top of HF (RPA@HF), both our TF result and the ACFDT result of Ref. [28]. Both results present a correct attractive behaviour without fake bumps, although the original HF interatomic potential (see Fig. 2) is repulsive. Nevertheless the dissociation curve is too flat (though less in our calculation than in Ref. [29]), and the binding energy is severely underestimated. However, the introduction of *GW* corrections introduces an important improvement also when starting from HF. The RPA@*GW*@HF curve (Fig. 2) is much closer to the accurate and the experimental result, though it presents a ∼10% overestimate of both the binding energy and the bond length. Finally Fig. 2 shows the curve calculated by solving the Bethe-Salpeter equation on top of a *GW* electronic structure and starting from HF. The result is, beyond any expectation, impressively good. The equilibrium length is within 0.02 bohr from the exact value, and the binding energy is overestimated by only 0.3 mHa (see Table 1). These values are even more accurate than the QMC value [17], outperforming quantum chemistry methods like MP2 and even CCSD as well [29, 56–58]. Notice also that any change we have introduced to the standard procedure, like solving the BSE directly on top of HF, or considering an unscreened kernel as in TDHF, immediately destroys the good agreement, even providing results that are much worse in some cases. Our BSE result seems better than the most advanced ACFDT TDDFT approximations of Ref. [27], like the RPA+ and RPA+SOSEX, which are unbound, and also the RPA+rSE, which provides a good binding energy and bond length but still presents a bump at 6–7 Bohr. The range-separated hybrid (RSH) functional of Ref. [29, 59, 60] is certainly the approximation that is closer to the physics underlying the *GW* approach in which correlations are addressed by the introduction of screening. RSH+MP2

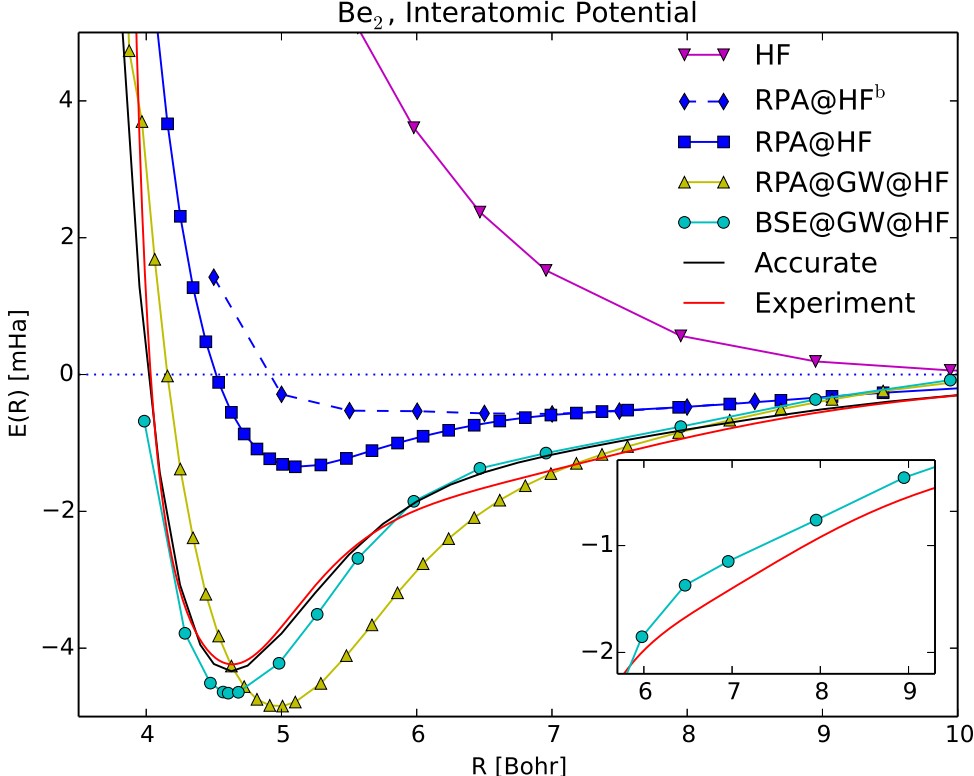

Figure 2: The interatomic potential of Be$_2$. Red line: experiment [1]; black line: accurate CI [12]; magenta line with triangle down symbols: HF; blue dashed line with diamonds: RPA on top of HF by ACFDT from Ref. [29]; blue line with squares: RPA on top of HF by TF; yellow line with triangle up symbols: RPA on top of *GW* and HF; cyan line with circles: BSE on top of *GW* and HF by the TF formula Eq. (6).

or RSH+RPAx tries to mimic this physics by introducing two ranges of different screened exchange. In Be$_2$ it obtains a clear improvement toward the good shape of the potential with no bump, but the binding energy and the bond length are not yet sufficiently reproduced. This can only be obtained by an approach presenting continuous variation of the screening at all ranges, like in GW+BSE.

Finally, the BSE on top of the *GW* approximation result even seems able to reproduce an unusual feature which has been pointed out in the accurate experiment of Ref. [1]. In the Inset of Fig. 2 we show a zoom on the region 6 to 9 bohr (∼3 to 4.5 Å) where van der Waals (vdW) interaction effects should enter into play and where, in the past, it was conjectured the existence of a vdW secondary (or even the only main) minimum. In this region the experimental curve presents an evident flattening which makes the interatomic potential deviate from the simple Morse potential, towards a more complex expanded Morse oscillator (EMO, see Ref. [1] and its Fig. 3). The EMO potential shape seems essential to best fit the experimental vibrational spectrum of Ref. [1] and seems a particularity of this dimer in which there is competition between covalent and vdW interactions. The vdW interaction is unable to produce a secondary minimum, but does have an influence on the shape of the interatomic potential, distorting it from the simple Morse oscillator towards a flattening. Our BSE@*GW*@HF curve also presents this flattening, though shifted with respect to experiment. It is nevertheless impressive that we were able to describe such a delicate balance between covalent and vdW bonding which

Table 1: $Be_2$ bond length $R_e$ and energy $D_e = E(R_e)$.

| Method | $R_e$ [bohr] | $D_e$ [mHa] |
|---|---|---|
| MP2 [29, 57] | 5.0 | -1.99 |
| CCSD [58] | 8.37 | -0.26 |
| CCSD(T) [55] | 4.64 | -3.00 |
| QMC DMC [17] | 4.65 | -2.82 |
| BSE@$GW$@HF by TF Eq. (1) | 4.65 | -4.66 |
| Accurate [12] | 4.63 | -4.34 |
| Experiment [1] | 4.64 | -4.24 |

only arises from correlations. This means that the Bethe-Salpeter equation provides an at least qualitatively good description of the high order correlation effects.

## 4  Conclusions

We have calculated the $Be_2$ interatomic potential along its dissociation path by an approach within *ab initio* many-body perturbation theory relying on the trace formula, as an alternative to the TDDFT ACFDT formalism. Our approach has been validated against TDDFT ACFDT calculations already presented in the literature. The introduction of *GW* corrections already largely improves the shape of the interatomic potential and also the binding energy and the bond length. At the level of the BSE we have obtained a $Be_2$ interatomic potential in very good agreement with experiment and with accurate CI calculations. We were even able to reproduce a flattening observed in the experimental potential which results from a delicate balance of correlations due to a competition between covalent and van der Waals bonding.

## Acknowledgments

This work is dedicated to the memory of János G. Ángyán, whom we acknowledge for inspiration and useful discussions. We thank Maria Hellgren for useful discussions and Ronald Cox for critical reading of our manuscript.

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
