# Peer review of "Ground-state correlation energy of beryllium dimer by the Bethe-Salpeter equation"

_SciPost Physics, doi:SciPost Phys. 8, 020 (2020)_

## Round 2 · Referee Report · Anonymous (Referee 1) · 2019-11-28

Strengths

1-formalism development
2-comparisons with state of the art techniques
3-explanations of the discrepancies

Weaknesses

1-Basis set dependency is missing
2-missing BSSE estimates

Report

The present article proposes to apply many-body derived schemes to estimate correlation energies in the difficult test-case of Be2 dimer dissociation. By comparing with experimental and very accurate results their different approximations, the authors propose that BSE+GW scheme is very accurate too. The formalism presentation is clear and the results convincing. I strongly recommend the present article suitable for publication. However I would be pleased if the authors could comment on the basis set dependency of their proposed scheme, as well as on their BSSE estimates. Even if they exclude it by following the recommendation of Baerends et al, it remains an important source of errors especially in the case of Be2, see Chem. Phys. Lett. 416, 370 (2005) for instance.
  • validity: top
  • significance: high
  • originality: high
  • clarity: top
  • formatting: good
  • grammar: excellent

Author:  Valerio Olevano  on 2020-01-16  [id 711]

(in reply to Report 1 on 2019-11-28)
Category:
answer to question

First of all, we thank the Referee 1 for the very high marks and positive appreciations of our work and paper.
We agree with the Referee that basis set convergence issues, as well as basis set superposition effects (BSSE), can be important sources of error, like both the Gerber and \'Angy\'an and also the Mentel and Baerends cited papers are showing. We modestly don't feel that tackling a study of BSSE and/or Counterpoise effects by our proposed GW+BSE and trace formula new approach can represent a general reference, like it is the case for the Mentel and Baerends study by CCSDT(T) and for the Gerber and \'Angy\'an by MP2 which are using much better established methodologies. In fact for such questions we fully relied on these previous authoritative studies. Our purpose here was only to compare, on as much as possible \underline{the very same footing}, our proposed new methodology with previous studies, the large majority of them having considered the Dunning cc-pV5Z as a reasonably converged basis set. So that this was also our choice.
However, upon request of both referees, we performed a basis set convergence study. The easiest and quickest is to carry on it within the time-dependent Hartree-Fock (TDHF) method, which differs from the BSE@GW@HF method only in the fact that the screened Coulomb interaction $W$ is replaced by the unscreened Coulomb interaction $v$ in both the calculation of single-particle energies and in the kernel. Notice also that our $GW$ calculation is self-consistent only on the energies, not on the wavefunctions that are kept at the level of HF in both TDHF and in GW+BSE.
In the joint figure we present the TDHF Be$_2$ interatomic potential using the cc-pVQZ vs the cc-pV5Z basis set, both for the cases considering the counterpoise (CP) correction and without. The lack of cc-pV6Z-RI auxiliary basis for our resolution-of-identity implemetation unfortunately does not allow us to increase further the cardinality of the basis. With regard to BSSE and the counterpoise correction, we find again the result already found by Mentel and Baerends, namely that the CP corrected result provides higher interaction energies (the non-CP corrected curves lie below) and so converges less quickly, though more smoothly. Then one can read the convergence of the cc-pV5Z with respect to the cc-pVQZ, which is $\sim$0.4-0.5 mHa, slightly larger than in CCSD(T) where it was found to rather be $\sim$0.2-0.3 mHa. We also checked the convergence on the GW+BSE result, which was found somehow larger, $\sim$0.6-0.8 mHa (depending if one reports the vertical difference or the difference between the minima). Notice however that calculations using the many-body methodology are not variational, so that increasing the number of elements in the basis set does not necessarily imply getting lower total energies. The trend between the QZ and the 5Z basis sets shows that, in contrast to TDHF, in GW+BSE a 6Z result should provide a lower binding energy which would be in the direction of the experiment and of the accurate CI calculation. Indeed, as one can check in Table I of our paper, our GW+BSE 5Z result overestimates the binding energy by 0.4 mHa.

Attachment:

basissetconv.pdf

---

## Round 2 · Referee Report · Anonymous (Referee 2) · 2019-12-19

Strengths

1 - original approach 2- interesting results

Weaknesses

1 - relation/equivalence between trace-formula and adiabatic-connection fluctuation-dissipation theorem formula not entirely clear

Report

This is a well-written manuscript presenting very interesting results for researchers working in the fields of electronic structure methods and simulations. The introduction sets clearly the problem, the approach and the key results obtained. The theory part is accurate and precise and points clearly the approximations that are introduced. The results part compares various level of approximations.
I'd like the authors to provide clarifications on the relation between TF and ACFDT. In the introduction, the authors refer to a work which shows the equivalence between the methods within RPA and TDDFT. However, in the Method part they contrast the in-principle exact ACFDT with the TF relying on the boson approximation- which seems to contradict the previous statement.
Also, it would be interesting if the authors could argue more on the stark difference due to the starting point. Is it possible to identify the shortcoming with the PBE one-particle solutions that causes the bump (e.g. self-interaction, double counting ...). I think this could be a useful addition.

Requested changes

1 - clarify apparent contradiction (see report)

The addition (see report) is desirable not requested. On the other hand, I think a more careful analysis on the effect of the basis set choice should be added as pointed in the other referee's report.

  • validity: high
  • significance: high
  • originality: high
  • clarity: high
  • formatting: good
  • grammar: excellent

Author:  Valerio Olevano  on 2020-01-16  [id 712]

(in reply to Report 2 on 2019-12-19)
Category:
answer to question

We also thank the Referee 2 for the high marks and positive appreciations.

For the analysis on the effect of the basis set choice: we invite the referee to check our answer to Referee 1.

On the relation between the TF and the ACFDT formulas: Our statements presented an apparent contradiction on this issue that we hope to have clarified in the modified version of the manuscript. The derivation of the ACFDT formula (under the assumption that the density does not change along the $\lambda$ AC path and is equal to the exact density, $\rho_\lambda(r) = \rho^\mathrm{exact}(r)$, which is always true for the electron gas homogeneous phase and in ACFDT TDDFT taking the DFT Kohn-Sham (KS) system as reference) does not rely on any approximation. In contrast, the derivation of the TF formula does rely on approximations, i.e. the quasi-boson approximation and the killing condition on the 0-order Slater determinant ground-state. Hence the ACFDT formula is in principle exact, whereas the TF is an approximated formula. However, the two formulas are equivalent within the RPA approximation. This has been demonstrated by Sawada, Bruekner et al. PR 108, 507 (1957) and PR 135 A392 (1964) for the case of the homogeneous electron gas (HEG); and by Furche J. Chem Phys. 129, 114105 (2008) in general for any system but taking the DFT Kohn-Sham as reference. This equivalence has not been demonstrated in the case of a non-zero exchange-correlation kernel beyond the RPA approximation. The difference between ACFDT and TF in beyond RPA correlation approaches is still to be explored. We notice that the ACFDT formula is exact only in the HEG case and within DFT-TDDFT, i.e. taking the exact density KS system as reference. ACFDT is not any more exact starting from other, e.g. HF or GW, reference systems. The fact that the density of the reference system is not kept constant along the $\lambda$ path (and exact), unlike KS, introduces additional terms in $\rho_\lambda$ [see e.g. Maggio and Kresse Phys. Rev. B 93, 235113 (2016)]. And these terms may not be negligible, as shown by Hesselmann & Gorling [Molecular Phys. 109, 2473 (2011)] for the case of HF as reference system.

On the identification of the shortcoming causing the unphysical bump in the RPA@PBE curve: As already noticed [Nguyen and Galli, J. Chem. Phys. 132, 044109 (2010)], the bump arises in RPA calculations on top of DFT approximations, that is LDA, GGA (PBE) and even hybrids like PBE0, but not on top of HF. In this work we found that interposing a $GW$ calculation, as in RPA@GW@PBE, suppresses the bump. In the joint figure we also show the HF and the EXX@PBE curve, as well as all incremental correlation contributions (dashed lines) to pass from an approximation to the next. That is, starting from the HF total energy $E_\mathrm{tot}^\mathrm{HF}$, which by definition neglects any correlation, we obtain all subsequent approximations by: $E_\mathrm{tot}^\mathrm{EXX@PBE} = E_\mathrm{tot}^\mathrm{HF} + E_\mathrm{corr}^\mathrm{EXX@PBE}$ $E_\mathrm{tot}^\mathrm{RPA@PBE} = E_\mathrm{tot}^\mathrm{EXX@PBE} + E_\mathrm{corr}^\mathrm{RPA@PBE}$ $E_\mathrm{tot}^\mathrm{RPA@GW@PBE} = E_\mathrm{tot}^\mathrm{RPA@PBE} + E_\mathrm{corr}^\mathrm{GW}$ We can argue that the bump in fact arises only from the EXX@PBE correlation contribution $E_\mathrm{corr}^\mathrm{EXX@PBE}$ (cyan dashed line in the joint figure). We can exclude the RPA@PBE correlation contribution $E_\mathrm{corr}^\mathrm{RPA@PBE}$ (red dashed line) as the origin of the bump: the magenta line, defined as the sum $E_\mathrm{tot}^\mathrm{HF} + E_\mathrm{corr}^\mathrm{RPA@PBE}$, does not present a bump. Finally, we can see that the $GW$ correlation contribution $E_\mathrm{corr}^\mathrm{GW}$ is effective in counteracting the spuriously generated EXX@PBE bump. Indeed, the full RPA@GW@PBE result, sum of all correlation contributions, does not present any more the bump. This is more impressing when noticing that our $GW$ is self-consistent only on eigenvalues, not on eigenfunctions which are kept at the PBE level. If the referee thinks that our explanation has really clarified the question of the unphysical bump, we can add it in an appendix to the paper, though this is somehow outside the main findings of our work. These reports are in any case public, so that interested people can always find them.

Attachment:

bumpstudy.pdf

Anonymous on 2020-01-19  [id 714]

(in reply to Valerio Olevano on 2020-01-16 [id 712])
Category:
answer to question

The authors'reply addressed thoroughly my remarks and I am happy for the current version to be published. I agree with the authors that, since this discussion will appear with the publication, there is no need for adding these side results to the manuscript.

---

## Round 3 · Author Response

Dear Editor,
we resubmit a new version of the manuscript following remarks by the Referees.
We in particular tried to clarify an apparent contradiction remarked by Referee 2 in the relation between ACFDT and Trace formulas.
And we better discuss the comparison with RSH methods, including the RSH+MP2 from a paper that was suggested by Referee 1.
Best regards.

---

## Round 3 · List of Changes

Page 1, last 6 lines: clarification of the ACFDT TF relationship.
Page 3, 1st column, penultimate paragraph: same issue.
Page 4, 2nd column, penultimate paragraph: widening of the discussion on RSH metthods including RSH+MP2.
Page 5: addition of two new references, [60] (pointed to us by Referee 1) and [61] (related to the same topic).

---

## Editorial Decision

published